# The epidemiology of soil-transmitted helminth infections in children up to 8 years of age: Findings from an Ecuadorian birth cohort

Irina Chis Ster[1], Hamzah F. Niaz[1], Martha E. Chico[2], Yisela Oviedo[2], Maritza Vaca[2], Philip J. Cooper[1,2,3]*

**1** Institute of Infection and Immunity, St George's University of London, London, United Kingdom, **2** Fundacion Ecuatoriana Para La Investigacion en Salud, Quito, Ecuador, **3** School of Medicine, Universidad Internacional del Ecuador, Quito, Ecuador

☯ These authors contributed equally to this work.
* pcooper@sgul.ac.uk

**Data Availability Statement:** All relevant data are within the manuscript and its Supporting Information files.

## Abstract

### Background

There are few prospective longitudinal studies of soil-transmitted helminth (STH) infections during early childhood. We studied the epidemiology of and risk factors for soil-transmitted helminth infections from birth to 8 years of age in tropical Ecuador.

### Methods

2,404 newborns were followed to 8 years of age with periodic stool sample collections. Stool samples were collected also from household members at the time of the child's birth and examined by microscopy. Data on social, environmental, and demographic characteristics were collected by maternal questionnaire. Associations between potential risk factors and STH infections were estimated using generalized estimated equations applied to longitudinal binary outcomes for presence or absence of infections at collection times.

### Results

Of 2,404 children, 1,120 (46.6%) were infected with at least one STH infection during the first 8 years of life. The risk of *A. lumbricoides* (16.2%) was greatest at 3 years, while risks of any STH (25.1%) and *T. trichiura* (16.5%) peaked at 5 years. Factors significantly associated with any STH infection in multivariable analyses included age, day-care (OR 1.34, 95% CI 1.03–1.73), maternal Afro-Ecuadorian ethnicity (non-Afro vs. Afro, OR 0.55, 95% CI 0.43–0.70) and lower educational level (secondary vs. illiterate, OR 0.31, 95% CI 0.22–0.45)), household overcrowding (OR 1.53, 95% CI 1.21–1.94)), having a latrine rather than a water closet (WC vs. latrine, OR 0.77, 95% CI 0.62–0.95)), and STH infections among household members (OR 2.03, 95% CI 1.59–2.58)). *T. trichiura* was more associated with

**Funding:** This work was supported by Wellcome Trust grants 074679/Z/04/Z (PJC) and 088862/Z/09/Z (PJC). The funders had no role in study design, data collection and analysis, decision to publish, or preparation of the manuscript.

**Competing interests:** The authors have declared that no competing interests exist.

poverty (high vs. low socioeconomic status, OR, 0.63, 95% CI 0.40–0.99)] and presence of infected siblings in the household (OR 3.42, 95% CI 2.24–5.22).

## Conclusion

STH infections, principally with *A. lumbricoides* and *T. trichiura*, peaked between 3 and 5 years in this cohort of children in tropical Ecuador. STH infections among household members were an important determinant of infection risk and could be targeted for control and elimination strategies.

## Author summary

Soil-transmitted helminths (STH) cause significant morbidity among children in low and middle-income countries (LMICs). We followed a birth cohort to 8 years of age in a rural area of coastal Ecuador and showed almost half acquired STH infections during childhood. The dominant STH parasites in this setting in children were *Ascaris lumbricoides* and *Trichuris trichiura* and infections peaked in frequency between 3 and 5 years of age. Risk of infection during childhood was associated with markers of marginalisation (Afro-Ecuadorian ethnicity and low educational level), household poverty (overcrowding and poor sanitation), and STH infections among other household members. There was evidence of a reduction in STH infection risk across all ages in study households over the calendar time course of this longitudinal study in parallel with improvements in living conditions, a period of economic growth, and increased government support for those living in extreme poverty. Our data indicate that targeting anthelmintic treatment at members of STH-infected households is likely to reduce the risk of infection in young children.

## Introduction

A quarter of humanity are at risk of infection with soil-transmitted helminths (STH), caused by *Ascaris lumbricoides*, *Trichuris trichiura*, and hookworm [1]. STH infections cause significant morbidity, particularly in children, through their effects on nutritional status, growth, and cognition [1, 2].

STH are infections of poverty and poor hygienic practices. Most current control programmes are focused on the provision of periodic anthelmintic treatments to reduce the numbers of children with high parasite burdens who are at greatest risk of morbidity. There is a need for more widespread implementation of integrated strategies that prevent infections through reducing exposures to infectious stages of STH parasites in the environment. However, such control programmes that include a multi-faceted approach of community health education for better hygiene practices, and improved access to clean water and sanitation, that together forms the WASH strategy [3], have had mixed results in endemic populations [3].

There are limited data on epidemiology of STH infections and risk factors for infection in pre-school through to school age children, especially from longitudinal studies. Most such studies with prospective follow-up have examined the impact of environmental and treatment interventions over relatively short periods (e.g. 12 months) [4–7]. To our knowledge, there are no previous studies addressing the epidemiology of STH infections from birth to school age and the role of the child's living environment in determining infection risk. We have shown previously in a birth cohort followed in an area of tropical Ecuador endemic for STH

infections that over 40% of children had at least one STH infection documented during the first 3 years of life and risk was determined by social factors such as maternal ethnicity and educational level [8]. Maternal STH infections also appeared to increase risk of infection in off-spring although it was not clear whether the increased risk was explained by a shared living environment or a specific maternal effect [8].

In the present study, we followed the same birth cohort to describe the epidemiology of STH infections during the first 8 years of life and identify individual and household factors affecting risk of infection during early childhood.

## Methods

### Ethics statement

The study protocol was approved by the ethics committees of the Hospital Pedro Vicente Maldonado (2005) and Universidad San Francisco de Quito (2010). Informed written consent was obtained from the child's mother and household members for collection of stool samples and data. The child was asked to provide minor assent at 8 years. Individuals with positive stools for STH infections were treated with a single dose of 400 mg albendazole if aged 2 years or greater and with pyrantel pamoate (11 mg/kg) if aged less than 2 years, according to Ecuadorian Ministry of Public Health recommendations [9, 10]. Pregnant women were offered treatment after the delivery of their child. All treatments were provided free by members of the study team.

### Study area and population

Detailed methodology of the study objectives, design, follow-up and sample and data collection for the ECUAVIDA birth cohort study are provided elsewhere [11]. Briefly, newborns whose families lived in the rural district of Quinindé, Esmeraldas Province, were recruited around the time of birth at the Hospital Padre Alberto Buffoni (HPAB) in the town of Quinindé between November 2005 and December 2009. The district is largely agricultural where the main economic activities relate to the cultivation of African palm oil and cocoa. The climate is humid tropical with temperatures generally ranging 23–32°C with yearly rainfall of around 2000-3000mm [12]. Inclusion criteria were being a healthy baby, collection of a maternal stool sample, and planned family residence in the district for at least 3 years [11].

### Study design and sample collection

Children were followed-up from birth to 8 years of age with data and samples collected at baseline during the initial home visit within 2 weeks of birth and at 3, 7, 13, 18, 24, 30 and 36 months and 5 and 8 years of age. Stool samples were also collected from mothers during the third trimester of pregnancy and from other household members during the initial home visit. Follow-ups were done either by scheduled visits to a dedicated clinic at HPAB or by home visits. At the initial home visit, a questionnaire was administered to the child's mother by a trained member of the study team to collect data on potential risk factors. The questionnaire included maternal and paternal data (age, ethnicity, education, occupation, and number of children), socio-economic data (number of material goods, monthly income, a household electrical connection, sources of drinking water), urban versus rural residence (defined by geographic location), number of sleeping rooms and number of people living in the household, and exposures to household pets, farming and farm animals [8]. Maternal questionnaires were repeated at 7 and 13 months and 2, 3, 5, and 8 years of age.

## Stool examinations

Stool samples were examined using four microscopic techniques to detect and/or quantify STH eggs and larvae [8] including direct saline wet mounts, formol-ether concentration, modified Kato-Katz, and carbon coproculture [8]. All stool samples were examined using all 4 microscopic methods where stool quantity was adequate. A positive sample was defined by the presence of at least one egg or larva from any of the above detection methods. Parasite burdens with *A. lumbricoides* and *T. trichiura* were quantified as eggs per gramme (epg) of stool using the results of the modified Kato-Katz method and categorized into light, moderate, and heavy intensities using WHO criteria [8] as follows: *A. lumbricoides* (light– 1–4,999; moderate– 5,000–49,999; and heavy—> = 50,000 epg) and *T. trichiura* (light– 1–999; moderate– 1000– 9,999; and heavy—> = 10,000 epg).

## Statistical analyses

Potential risk factors considered included individual (for participant child—gender, age, birth order, breast feeding duration, day care, and anthelmintic treatments), parental (age, educational status, ethnicity, allergic symptoms, atopy, and STH infections), and household (area of residence, socioeconomic status, overcrowding, monthly income, construction materials, material goods [fridge, TV radio, and HiFi], potable water, type of bathroom, pets, pigs [putative risk for *Ascaris* spp. infections], agricultural activities, and STH infections in household members). Socioeconomic variables were combined to create a socio-economic status (SES) index by using principal components analysis for categorical data as described [8].

To estimate age-dependent risk of STH infections around the time of birth of cohort children, using data from stool samples collected from members of cohort households that have a clustered hierarchical structure (defined by households and individual participants), we used two-level logistic regression (households at the first and participants within households at the second levels) [13] to account for intra- and inter-household variability. Estimates were based on 6,800 stool samples collected from 1,973 households with a median size of 3 members (interquartile range, 1–5). The approach produced adequate associated standard errors which accounted for the dependencies in the data attributable to participants living within the same household.

The presence or absence of STH infections (any or *A. lumbricoides* or *T. trichiura*) from birth to 8 years were defined as a longitudinal binary outcome. Stool sample collections were done periodically as described with some time variation although for analyses purposes these were considered fixed. We used generalized estimation equations models (GEE) to fit population-averaged models [14] to understand the effects of age, childhood, parental and household characteristics, and household STH infections (any member, parents, any excluding parents, and siblings only) on the age-dependent risk of acquiring STH infections in childhood. Binary random effects models were also considered [13, 14] as described [15]. The GEE approach provides population-average while random effects analysis gives subject-specific estimates. The latter was particularly appropriate for time-varying exposures (i.e. bathroom type, anthelmintic treatments, and household pigs) and can indicate associations of outcome with individual changes in these exposures during follow-up. We used the assumption of an unstructured correlation structure [13–15] and, for the sake of simplicity and given that our questions were addressed at the population level, we have presented and commented on population average estimates only. Associations and their uncertainties were measured by odds ratios (OR) and their corresponding 95% confidence intervals (CIs). ORs derived from these longitudinal models estimated associations between potential explanatory variables and the age-dependent risk of STH outcomes. The interpretation is similar to a cross-sectional OR. Minimally

adjusted models (for age in polynomial forms to the power of 5) investigated the sole association of each factor on STH outcomes. Multivariable models were subsequently built using variables with P<0.1 in minimally adjusted models and the smallest associated quasi-likelihood value under the independence model criterion (QIC) for GEEs [15, 16] on a complete data sample. The QIC criterion is an adaptation of the AIC (Akaike's information criterion) criterion for GEEs for model choice [16, 17]. The final most parsimonious model using a complete data sample was fit back to the original data on as many observations as possible. Because longitudinal cohorts are subject to attrition at follow-up, we analysed patterns in missing data for any STH infection and did sensitivity analyses. GEE estimations were based on the missing completely at random assumption [15, 18]. Random effects, based on maximum likelihood estimation, were also fit under missing at random assumption [15, 18] and did not produce very different estimates in terms of ORs and standard errors [15, 16]. Statistical significance was inferred by P<0.05. Predictions for age-dependent risk of STH, *A. lumbricoides* and *T. trichiura* infections were displayed against raw data. Distributions of infection intensities (epg) for *A. lumbricoides* and *T. trichiura* were highly skewed and analysed using zero-inflated models [19] assuming two sources of zeros: those that partly belonged to the count distribution with zeros resulting from undetectable infections and those partly representing a negative test. Estimates of effect derived from these analyses indicated associations of "non-zero occurrences" (provided as ORs) or fold-change (in epg) with covariates. All estimates were age-adjusted and accounted for the hierarchical structure of outcomes. All statistical analyses were done using Stata version 16 [20] and raw data used for the analyses are provided (S1 Data).

## Results

### Study participants

Of 2,404 newborns recruited, 98.2% had at least one stool sample examined during follow-up between 3 months and 8 years (median 6 samples (IQR 4–7) from a maximum of 9 follow-up time points for stool sampling). A flow diagram showing observation times, losses to follow-up, and samples analysed is provided in Fig 1. The period from the recruitment of the first child to follow up at 8 years of the last child was from 18[th] November 2005 to 23[rd] November 2017.

### Epidemiology of STH Infections in study children and their households

Cumulative proportions of infections to 8 years were 46.6% for infections with any STH parasite, 37.5% for *A. lumbricoides*, 26.0% for *T. trichiura*, 1.3% for hookworm, and 1.6% for *S. stercoralis*. Corresponding values for other parasites were: *Hymenolepis nana* 4.3%, *Hymenolepis diminuta* 0.3%, *Enterobius vermicularis* 0.8%, and *Taenia* spp. 0.1%. The proportions infected with any STH infection, *A. lumbricoides*, and *T. trichiura* at each sampling time to 8 years is shown in Fig 2, as is STH prevalence in mothers, fathers, and in any household member around the time of birth of the child. Peak STH prevalence was seen at 3 years for *A. lumbricoides* (16.2%) and at 5 years for any STH (25.1%) and *T. trichiura* (16.5%). *A. lumbricoides* was the first STH parasite to be acquired in infancy but by 8 years of age *T. trichiura* prevalence was comparable to that of *A. lumbricoides*. Prevalence among mothers (any STH 46.2%, *A. lumbricoides* 27.0%, and *T. trichiura* 28.7%), fathers (any STH 27.2%, *A. lumbricoides* 14.1%, and *T. trichiura* 16.7%), and households (any household member infected: any STH 59.5%, *A. lumbricoides* 38.7%, and *T. trichiura* 43.6%) were greater than the peak STH prevalence in cohort children. Relatively few children had heavy parasite burdens with *A. lumbricoides* and very few had heavy burdens with *T. trichiura* (Fig 3): proportions of infected children with heavy parasite burdens peaked at 24–30 months for *A. lumbricoides* (9%). The age-dependent

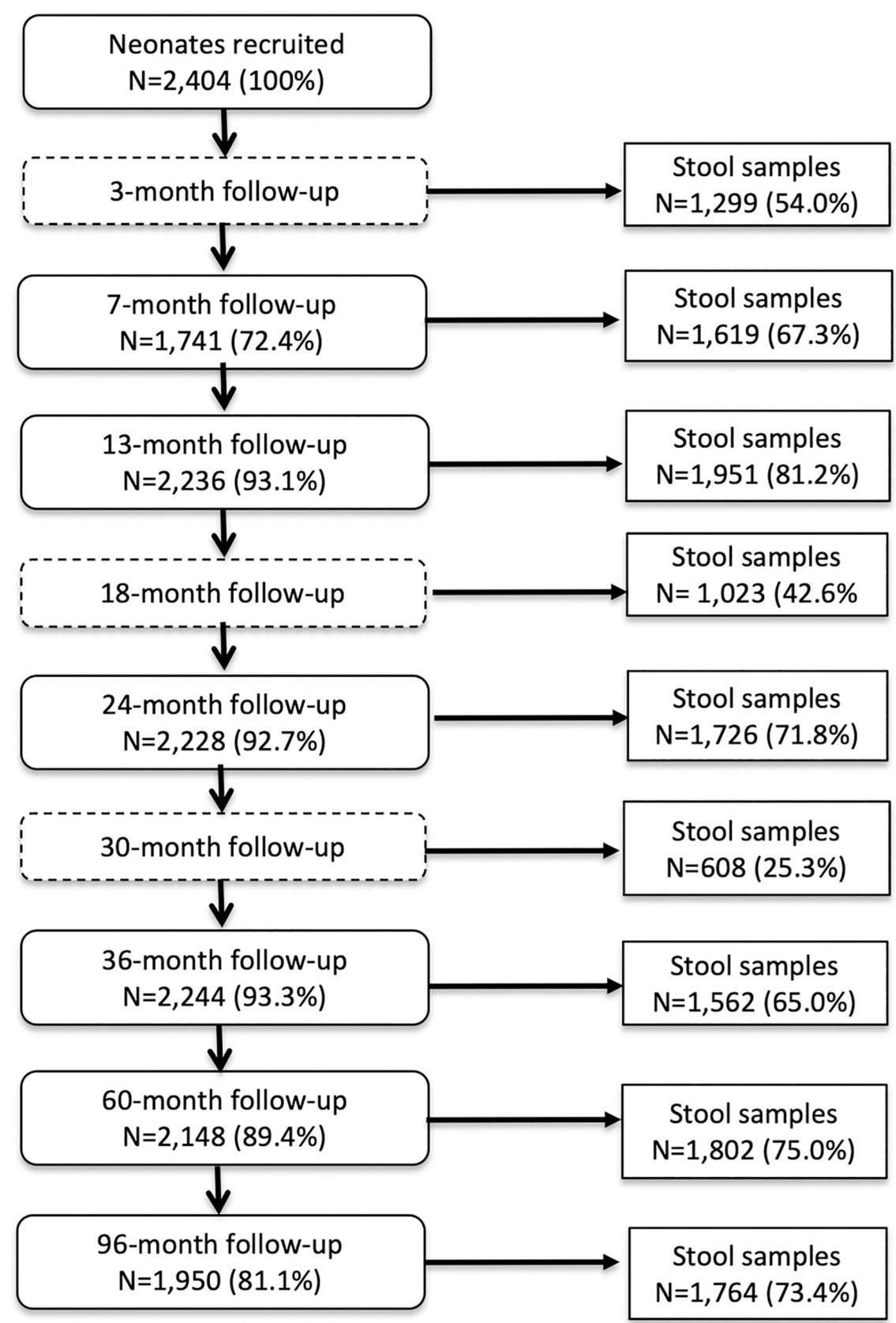

**Fig 1. Flow diagram to show follow-up of cohort to 8 years and stool sampling.** Denominator for all proportions is 2,404. A child with any stool sample collected during follow-up was included in the analysis.

estimated risk of STH infections among 6,800 household members of 1,973 households sampled around the time of birth of the child are shown in Fig 4 and S1 Table: a polynomial shaped age-prevalence profile was observed with peak prevalence for any STH infection at around 13 years (42%, 95%CI 38%-45%), at 12 to 14 years for *A. lumbricoides* (16.6%, 95%CI 14.3%-18.8%) and at 11–12 years (27%, 95%CI 24%-30%) for *T. trichiura*.

## Factors associated with STH Infections in longitudinal analyses

Age-dependent prevalence estimates for STH infections predicted by the longitudinal models against observed proportions are shown in Fig 5. Predictions were reasonably close to observed values (S2 Table)–the observed values were included in the 95% CIs of the predictions. Distributions of individual, parental, and household characteristics for the 2,404 cohort children and age-adjusted and multivariable associations of these factors with any STH infection in childhood are shown in Table 1. In age-adjusted analyses (Table 1), any STH infection was positively associated with: a) individual—increasing birth order and no recent anthelmintic treatment; parental–maternal and paternal Afro-Ecuadorian ethnicity and lower educational level; household–lower socioeconomic status, overcrowding, lower household income, construction using traditional materials, fewer material goods, and having a latrine rather than a WC. With respect to STH infections in other household members at the time of the child's birth, any STH infection was associated with maternal, paternal, and sibling infections. In

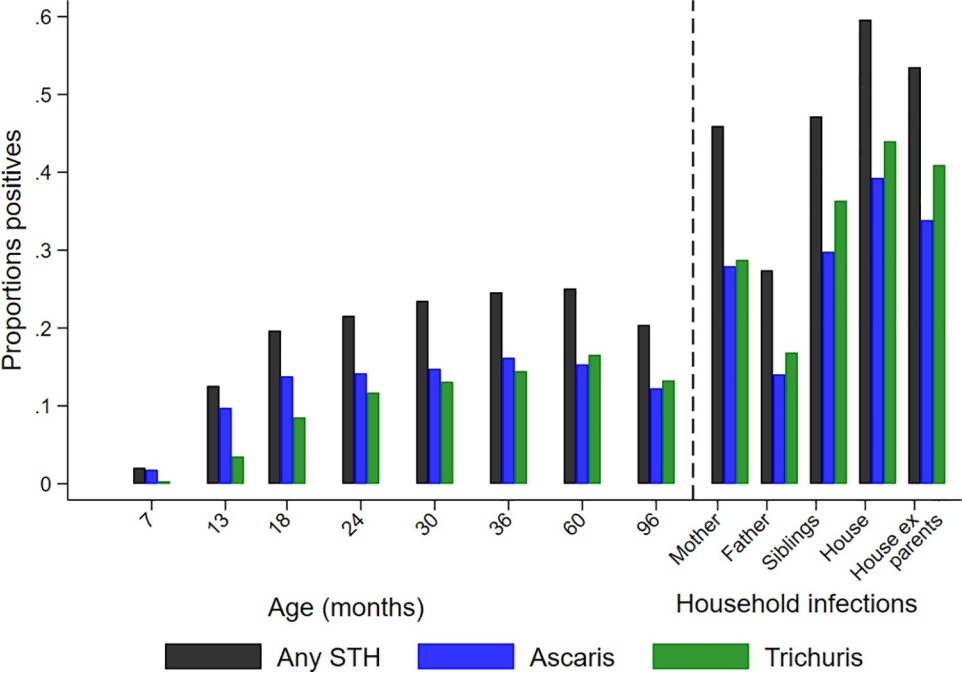

**Fig 2. Prevalence of soil-transmitted helminth (STH) infections observed in children sampled at each age and among the child's mother, father, siblings, any household member (house), and any household member excluding (ex) parents.** Bars represent any STH infection (black), *A. lumbricoides* (blue), and *T. trichiura* (green). Samples for cohort are shown in Fig 1 and for mothers (2,390), fathers (997), and households (1,971), siblings (1,051) and households except parents (1,344).

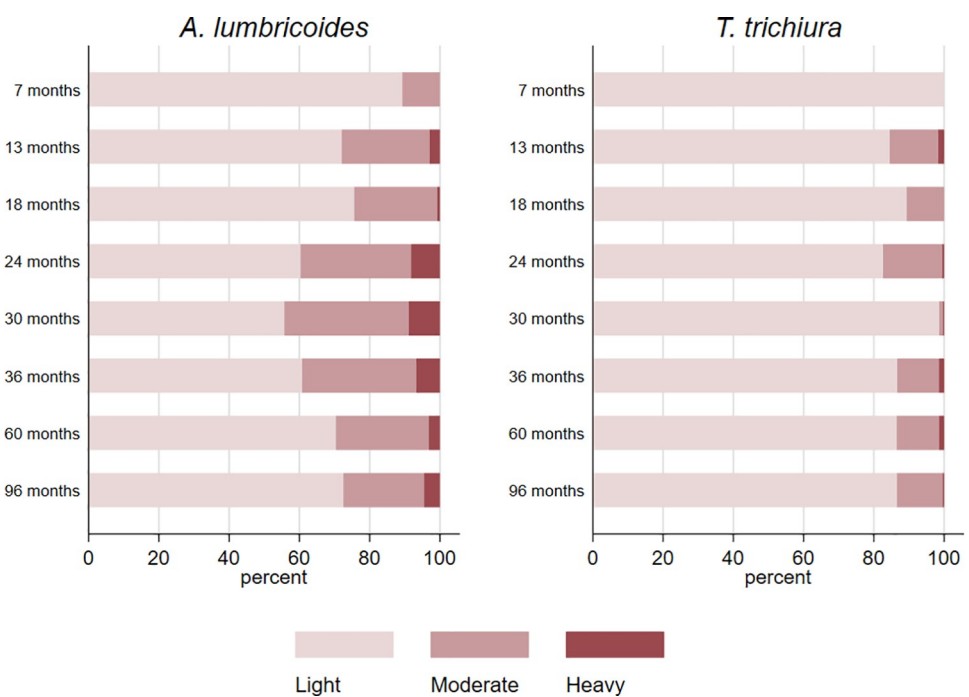

**Fig 3. Relative frequencies of parasite burdens (light, moderate, and heavy) with *A. lumbricoides* and *T. trichiura* observed among infected children during the first 8 years of life.** Infection intensity categories were as recommended by WHO: A. lumbricoides, light (1–4,999), moderate (5,000–49,999), heavy (> = 50,000); *T. trichiura*, light (1–999), moderate (1,000–9,999), heavy (> = 10,000)

multivariable analyses, factors associated with any STH infection were: a) individual–increasing age and any day-care attendance during the first 3 years of life (OR 1.34, 95% CI 1.03–1.73); b) maternal–Afro-Ecuadorian ethnicity (non-Afro vs. Afro, OR 0.55, 95% CI 0.43–0.70) and lower educational level (vs. illiterate, primary OR 0.48 [95% CI 0.35–0.65] and secondary OR 0.31 [95% CI 0.22–0.45]); and c) household–household overcrowding (OR 1.53, 95% CI 1.21–1.94) and bathroom type (WC vs. latrine, OR 0.77, 95% CI 0.62–0.95); and having any household member with an STH infection (OR 2.03, 95% CI 1.59–2.58). The latter variable was selected for inclusion in the multivariable model because it showed best fit in QIC analyses on the subsample with data on STH infections for the mother, father, and at least one sibling (Table 2).

Age-adjusted and multivariable associations for risk factors with *A. lumbricoides* and *T. trichiura* are shown in S3 and S4 Tables, respectively. Similar associations were observed in age-adjusted analyses as seen for any STH infections. However, anthelmintic treatment was significantly inversely associated with *A. lumbricoides* (OR 0,73, 95% CI 0.60–0.89, P = 0.001), while paternal allergy (OR 0.47, 95% CI 0.24–0.95, P = 0.036) and having a WC as bathroom (OR 0.67, 95% CI 0.54–0.83, P<0.001) were

Models using age-adjusted variables with lowest QIC values were considered to provide best fit using complete data from 693 households with 4,287 observations for which complete data were available for all variables under consideration. The sub-sample used for analysis did not differ from the rest of cohort with respect to variables shown in Table 1 except for overcrowding inversely associated with *A. lumbricoides* but not *T. trichiura* infections; household STH infections were associated with both infections (any member with *A. lumbricoides*, OR 2.56, 95% CI 2.02–3.32, P<0.001; and with *T. trichiura* OR 3.57, 95% CI 2.59–4.92, P<0.001),

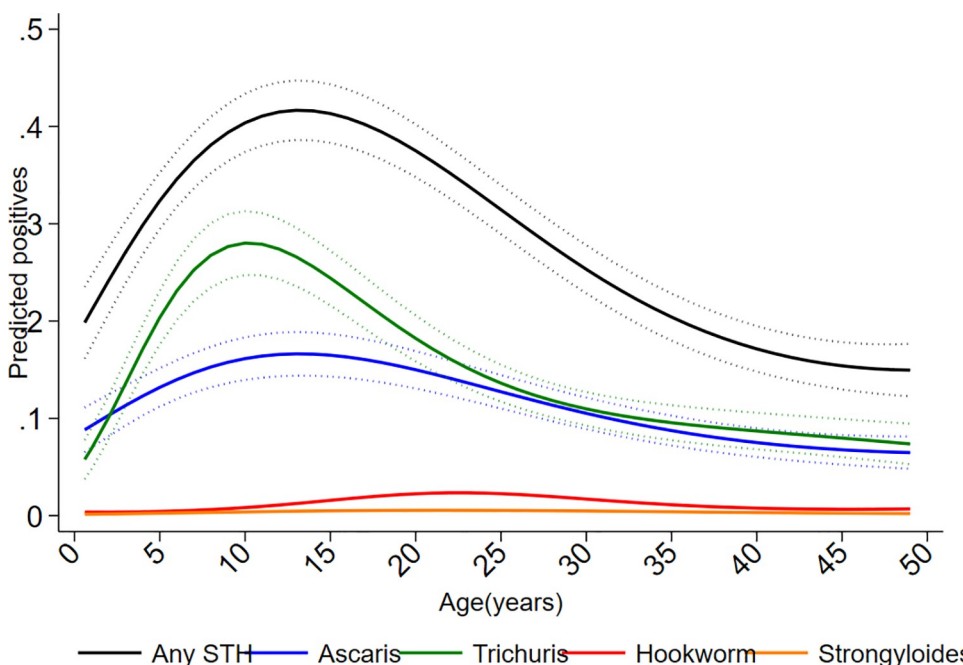

**Fig 4. Age-dependent risk of infection with any soil-transmitted helminth (STH) and individual STH parasites inferred from household data collected at the time of recruitment of the ECUAVIDA cohort.** Age-dependent proportions (and 95% confidence intervals) of positives were predicted using two-level logistic regression fitted to a binomial response accounting for household clustering. Age modelled using a 5th degree polynomial. The data consist of samples collected from household members at the time of recruitment of the cohort.

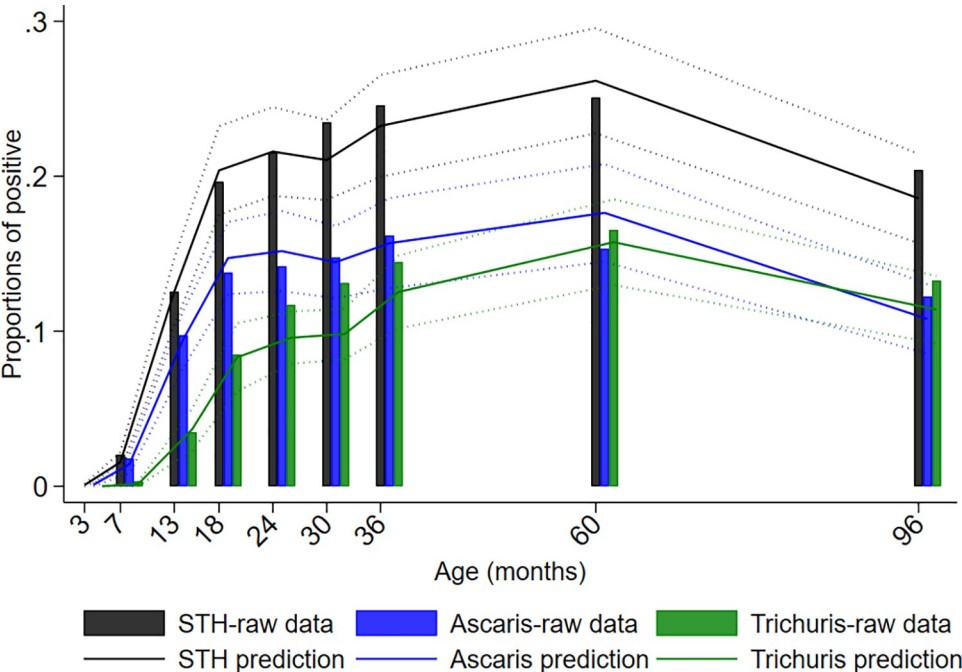

**Fig 5. Observed proportions and predicted age-dependent prevalence of infection with any soil-transmitted helminth (STH).** *A. lumbricoides* (Ascaris), and *T. trichiura* (Trichuris) in the cohort during the first 8 years of life. Age is shown in months.

**Table 1. Age-adjusted and multivariable associations between any soil-transmitted helminth (STH) infection during first 8 years of life and individual, parental, and household factors including STH infections among household members.**

| VARIABLES | CATEGORY | FREQUENCIES | | AGE-ADJUSTED ONLY | | | | MULTIVARIABLE | | | |
|---|---|---|---|---|---|---|---|---|---|---|---|
| | | NO | (%) | OR | P-value | 95%CI LOW | 95%CI HIGH | OR | P-value | 95%CI LOW | 95%CI HIGH |
| **CHILDHOOD FACTORS** | | | | | | | | | | | |
| Age | One-month effect | | | **4.073** | **<0.001** | **2.837** | **5.847** | **3.656** | **<0.001** | **2.478** | **5.395** |
| $Age^2$ | | | | **0.925** | **<0.001** | **0.903** | **0.948** | **0.931** | **<0.001** | **0.907** | **0.957** |
| $Age^3$ | | | | **1.002** | **<0.001** | **1.001** | **1.003** | **1.002** | **<0.001** | **1.001** | **1.003** |
| **Sex** | Male | 1228 | (51.1%) | | | | | | | | |
| | Female | 1176 | (48.9%) | 1.038 | 0.753 | 0.825 | 1.306 | | | | |
| **Birth order** | 1st-2nd | 1195 | (49.7%) | **1** | | | | | | | |
| | 3rd-4th | 746 | (31.0%) | **1.396** | **0.003** | **1.119** | **1.741** | | | | |
| | >= 5th | 463 | (19.3%) | **2.461** | **<0.001** | **1.800** | **3.364** | | | | |
| **Breastfeeding (months)** | 0–6 | 232 | (9.7%) | 1 | | | | | | | |
| | 7–12 | 956 | (39.8%) | 0.904 | 0.606 | 0.615 | 1.328 | | | | |
| | >12 | 1023 | (42.6%) | 0.803 | 0.234 | 0.559 | 1.153 | | | | |
| **Day care to 3 years** | No | 1969 | (81.9%) | 1 | | | | | | | |
| | Yes | 435 | (18.1%) | **1.373** | **0.007** | **1.089** | **1.732** | **1.337** | **0.027** | **1.034** | **1.729** |
| *Recent treatment‡ | 0–1 | 96 | (8.1%) | **1** | | | | | | | |
| | 2–3 | 1059 | (44.1%) | **0.782** | **0.012** | **0.647** | **0.947** | | | | |
| | >= 4 | 1149 | (47.8%) | | | | | | | | |
| **MATERNAL FACTORS** | | | | | | | | | | | |
| Age (years) | <= 20 | 644 | (26.8%) | 1 | | | | | | | |
| | 21–29 | 1161 | (48.3%) | 1.156 | 0.284 | 0.887 | 1.507 | | | | |
| | >= 30 | 599 | (24.9%) | 0.924 | 0.571 | 0.705 | 1.213 | | | | |
| **Ethnicity** | Afro-Ecuadorian | 616 | (25.6%) | **1** | | | | | | | |
| | Non-Afro-Ecuadorian | 1788 | (74.4%) | **0.512** | **<0.001** | **0.410** | **0.639** | **0.551** | **<0.001** | **0.431** | **0.704** |
| **Educational status** | Illiterate | 368 | (15.3%) | **1** | | | | | | | |
| | Primary | 1412 | (58.7%) | **0.462** | **<0.001** | **0.345** | **0.619** | **0.478** | **<0.001** | **0.351** | **0.651** |
| | Secondary | 624 | (26.0%) | **0.229** | **<0.001** | **0.167** | **0.314** | **0.314** | **<0.001** | **0.219** | **0.450** |
| **Allergic symptoms** | No | 2269 | (94.4%) | 1 | | | | | | | |
| | Yes | 119 | (5.0%) | 0.706 | 0.131 | 0.449 | 1.110 | | | | |
| **Atopy** | No | 1472 | (71.7%) | 1 | | | | | | | |
| | Yes | 582 | (28.3%) | 0.908 | 0.471 | 0.698 | 1.181 | | | | |
| **PATERNAL FACTORS** | | | | | | | | | | | |
| Age (years) | <= 20 | 244 | (10.2%) | 1 | | | | | | | |
| | 21–29 | 1087 | (45.2%) | 0.835 | 0.33 | 0.581 | 1.200 | | | | |
| | >= 30 | 1073 | (44.6%) | 0.964 | 0.847 | 0.661 | 1.405 | | | | |
| **Ethnicity** | Afro-Ecuadorian | 541 | (23.2%) | **1** | | | | | | | |
| | Non-Afro-Ecuadorian | 1789 | (76.8%) | **0.567** | **<0.001** | **0.450** | **0.714** | | | | |
| **Educational status** | Illiterate | 341 | (15.5%) | **1** | | | | | | | |
| | Primary | 1170 | (53.2%) | **0.517** | **<0.001** | **0.366** | **0.731** | | | | |
| | Secondary | 689 | (31.3%) | **0.364** | **<0.001** | **0.248** | **0.535** | | | | |
| **Allergic symptoms** | No | 2138 | (96.4%) | 1 | | | | | | | |
| | Yes | 80 | (3.6%) | 0.670 | 0.205 | 0.360 | 1.246 | | | | |
| **Atopy** | No | 786 | (64.7%) | 1 | | | | | | | |
| | Yes | 429 | (35.3%) | 1.054 | 0.749 | 0.763 | 1.457 | | | | |
| **HOUSEHOLD FACTORS** | | | | | | | | | | | |
| **Area of residence** | Urban | 1685 | (70.1%) | 1 | | | | | | | |

*(Continued)*

**Table 1.** (Continued)

| VARIABLES | CATEGORY | FREQUENCIES | | AGE-ADJUSTED ONLY | | | | MULTIVARIABLE | | | |
|---|---|---|---|---|---|---|---|---|---|---|---|
| | | NO | (%) | OR | P-value | 95%CI LOW | 95%CI HIGH | OR | P-value | 95%CI LOW | 95%CI HIGH |
| | Rural | 719 | (29.9%) | 0.996 | 0.974 | 0.778 | 1.276 | | | | |
| Socio-econ status | Low | 797 | (33.2%) | 1 | | | | | | | |
| | Medium | 800 | (33.3%) | 0.750 | 0.04 | 0.570 | 0.988 | | | | |
| | High | 807 | (33.6%) | 0.463 | <0.001 | 0.356 | 0.602 | | | | |
| House overcrowding | <3 | 1342 | (55.8%) | 1 | | | | | | | |
| | >=3 | 1062 | (44.2%) | 2.096 | <0.001 | 1.676 | 2.622 | 1.532 | <0.001 | 1.212 | 1.936 |
| Monthly income | <1 | 1379 | (64.6%) | 1 | | | | | | | |
| | >=1 | 755 | (35.4%) | 0.850 | 0.005 | 0.759 | 0.952 | | | | |
| Household construction | Wood/bamboo | 615 | (25.9%) | 1 | | | | | | | |
| | Cement/brick | 1764 | (74.2%) | 0.622 | <0.001 | 0.484 | 0.799 | | | | |
| Material goods | 0–2 | 1199 | (49.9% | 1 | | | | | | | |
| | 3–4 | 1205 | (50.1%) | 0.665 | <0.001 | 0.534 | 0.827 | | | | |
| Potable water | No | 1579 | (65.7%) | 1 | | | | | | | |
| | Yes | 825 | (34.3%) | 1.109 | 0.355 | 0.891 | 1.381 | | | | |
| *Bathroom (birth) | Latrine | 1687 | (70.2%) | 1 | | | | | | | |
| | WC | 717 | (29.8%) | 0.719 | <0.001 | 0.600 | 0.861 | 0.768 | 0.017 | 0.619 | 0.953 |
| *Dog in house(birth) | No | 2056 | (85.5%) | 1 | | | | | | | |
| | Yes | 348 | (14.5%) | 1.014 | 0.923 | 0.764 | 1.346 | | | | |
| *Cat in house (birth) | No | 2020 | (85.0%) | 1 | | | | | | | |
| | Yes | 384 | (15.0%) | 1.122 | 0.511 | 0.796 | 1.581 | | | | |
| *Pigs (birth) | No | 1691 | (70.3%) | 1 | | | | | | | |
| | Yes | 713 | (29.7%) | 1.056 | 0.613 | 0.855 | 1.305 | | | | |
| Agriculture | No | 1159 | (48.2%) | 1 | | | | | | | |
| | Yes | 1245 | (51.8%) | 0.865 | 0.213 | 0.689 | 1.087 | | | | |
| HOUSEHOLDS STH INFECTIONS | | | | | | | | | | | |
| Any in household | No | 796 | (40.4%) | 1 | | | | | | | |
| | Yes | 1175 | (59.6%) | 2.721 | <0.001 | 2.134 | 3.471 | 2.026 | <0.001 | 1.592 | 2.577 |
| Mother | No | 1292 | (54.1%) | 1 | | | | | | | |
| | Yes | 1098 | (45.9%) | 3.105 | <0.001 | 2.507 | 3.846 | | | | |
| Father | No | 724 | (72.6%) | 1 | | | | | | | |
| | Yes | 273 | (27.4%) | 2.370 | <0.001 | 1.637 | 3.433 | | | | |
| Any except parents | No | 625 | (46.5%) | 1 | | | | | | | |
| | Yes | 719 | (53.5%) | 3.015 | <0.001 | 2.306 | 3.942 | | | | |
| Siblings | No | 555 | (52.8%) | 1 | | | | | | | |
| | Yes | 496 | (47.2%) | 3.315 | <0.001 | 2.472 | 4.446 | | | | |

Estimates show population-averaged estimates using generalized estimating equations. Missing STH variables: STH in household (n = 433), mother (14), father (1,407), any except parents (1,060), and siblings (1,353).

*Time-varying variables. Birth–data collected around the time of birth of the child.

‡Anthelmintic treatment during the previous 12 months. Frequencies show number of anthelmintic treatments received during follow-up. N-Afro.–non-Afro-Ecuadorian; Prim.–primary completed; Second.–secondary completed; SES–socioeconomic status; overcrowding–persons/sleeping room; Income- monthly household income as number of basic salaries of US$472; Non-trad.–non-traditional (wall construction with bricks/cement/blocks); trad.–traditional (wall construction with wood/bamboo); material goods–number of household electrical goods; pigs–keeping pigs around the house; agriculture–child lives on a farm or visits a farm at least once a week.

**Table 2. Soil-transmitted helminth infections (STH) among household members showing best age-adjusted model fitted by generalized estimating equation models and using quasi-likelihood under the independence model criterion (QIC).**

| Household member infected with STH | Any STH | A. lumbricoides | T. trichiura |
|---|---|---|---|
| Any household member | **3050.281** | **2429.737** | 1863.549 |
| All except parents | 3081.447 | 2461.389 | 1854.234 |
| Siblings | 3091.519 | 2457.483 | **1845.065** |
| Mother | 3084.909 | 2460.072 | 1913.345 |
| Father | 3113.232 | 2450.395 | 1931.325 |

particularly having a mother with greater parasite burdens (vs. uninfected, *A. lumbricoides* OR 6.07, 95% CI 4.37–8.43, P<0.001; *T. trichiura* OR 18.26, 95% CI 9.14–36.46, P<0.001) and *T. trichiura* infections among siblings (OR 5.09, 95% CI 3.23–8.03, P<0.001). Associations of infection intensities with *A. lumbricoides* and *T. trichiura* in age-adjusted models are shown in S5 and S6 Tables, respectively: findings for both infections were consistent with those of analyses for binary outcomes.

In multivariable analyses, similar conclusions were inferred as for any STH infection except *T. trichiura* infection that appeared to be more associated with greater poverty (high vs. low SES, OR 0.63, 95% CI 0.40–0.99, P = 0.047) and *T. trichiura* infections among siblings (OR 3.42, 95% CI 2.34–5.22, P<0.001) rather than any household member.

Population-average estimates did not differ qualitatively from subject-specific effects for the time-varying exposures, anthelmintic treatment, bathroom type, and domestic pigs (S7 Table) —but interpretation relates to individual changes. Sensitivity analyses using extreme values for missing data for any STH infections over the period of follow-up (i.e. one in which the missing responses were either considered all negative or all positive) generally preserved the sense of associations observed for analyses using complete data (S8 Table): age-adjusted estimates for these extreme scenarios are displayed beside complete data estimates in S8 Table. Estimates from these extreme value analyses were comparable to complete data estimates (i.e. 95% CIs intersected) indicating robustness of our findings.

## Temporal trends in household determinants of STH risk

The risk of STH infections was greater in children living in households at the time of the child's birth compared to the risk in cohort children after 3 years of age (Figs 2, 4 and 6, and S1 Table). For example, estimated risk in siblings versus cohort children at 8 years were: any STH 38.1% vs. 20.4%, *A. lumbricoides* 15.3% vs. 12.2%, *T. trichiura* 25.0% vs. 13.3%). This apparent decline in STH risk, representing an effect primarily on risk of *T. trichiura* infection, could be explained by environmental changes in the District occurring over the 12-year calendar period (2006 to 2017) during which the study was done. Environmental determinants of STH infections in the cohort in age-adjusted analyses included household overcrowding, living in a house constructed with traditional materials, and type of bathroom (any STH but not *T. trichiura*) (Table 1). Data on overcrowding, household construction (traditional vs. non-traditional), and type of bathroom (WC vs. latrine) were collected periodically by questionnaire and empirical trends (over calendar time) of changes in relative frequencies of households with these attributes are shown in Fig 7. Between 2006 and 2017 there were large decreases in proportions of study households with overcrowding (from 64.3% to 30.8%) and use of traditional construction materials (34.3% to 15.1%), while those with a WC increased (from 24.4% to 91.5%). Frequencies of households with large farm animals, with no evidence for an association with STH infections in this analysis, showed a less clear trend.

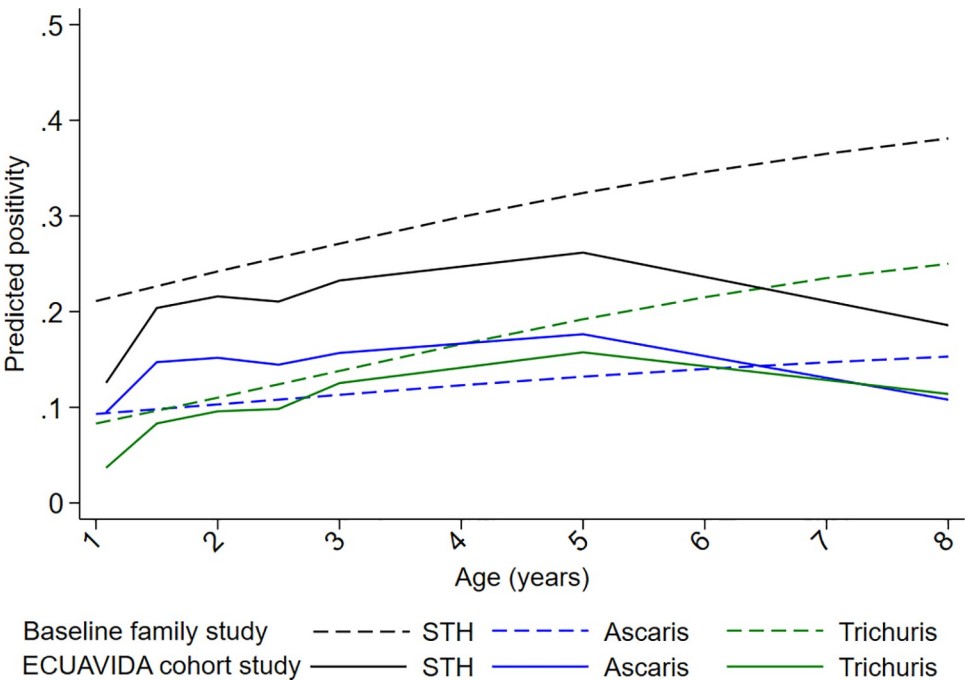

**Fig 6. Age-dependent prevalence of infections with soil-transmitted helminths (STH).** This includes any STH, *A. lumbricoides* (Ascaris), and *T. trichiura* (Trichuris) between 1 and 8 years of age comparing ECUAVIDA cohort (solid lines) with risk at the time of cohort recruitment among other children in the household (Baseline family study–dotted lines).

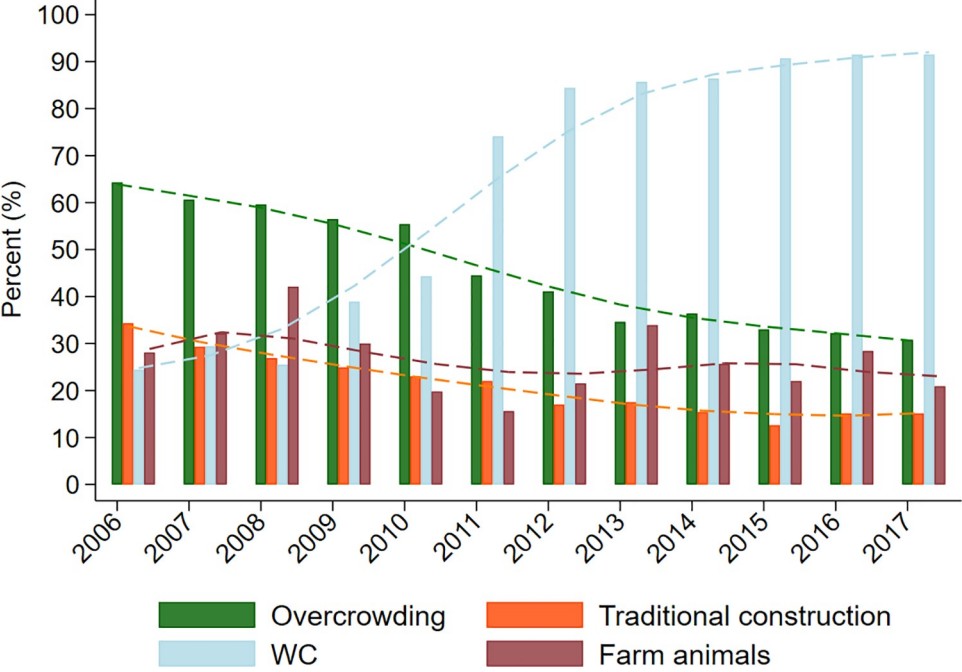

**Fig 7. Temporal changes in household characteristics during recruitment and follow-up of cohort between 2006 and 2017.** Overcrowding—> = 3 persons /sleeping room; traditional construction–wood and or bamboo walls; WC–water closet connected to municipal sewage or septic tank; farm animals–any large farm animal around the household. Number of households providing data by year: 2006 (431 households); 2007 (996); 2008 (1709); 2009 (2229); 2010 (1893); 2011 (1707); 2012 (1019); 2013 (841); 2014 (623); 2015 (540); 2016 (571); and 2017 (578). Household data were collected when the child was aged 0, 7, 13, 24, 36, 60, and 96 months and could provide data a maximum of once in any calendar year. Poly.—empirical polynomial lowess smoothing with suggestive trends across calendar time.

## Discussion

We studied the epidemiology of and risk factors for STH infection during the first 8 years of life in a birth cohort from a largely rural District in tropical coastal region of Ecuador. Almost half of the children had at least one STH infection documented during the first 8 years of life, almost exclusively with *A. lumbricoides* and *T. trichiura*, and few children harboured moderate to heavy parasite burdens with either parasite. Several individual, maternal, and household factors indicative of greater poverty and social marginalisation were associated with STH infection risk. Having another household member infected with STH around the time of the child's birth, particularly having a mother with moderate to high parasite burdens, was a strong risk factor for childhood STH. There was evidence of temporal changes in STH risk independent of age during the study.

Strengths of this study were the large sample size and longitudinal design with repeated sampling of cohort children to ascertain STH infection status. We used longitudinal repeated-measures analyses to model risk of infection during the first 8 years of life, an approach that is more informative and potentially improves the precision of estimates compared to more traditional (cross-sectional) methods. This analytic approach allows insights into the STH infection dynamics from birth to school age by deriving age-dependent prevalence of infection in the population and accounting for the temporal dependencies and hierarchical structure of the data while considering all complete observations in a unified manner thus minimizing the loss of information. We collected detailed information on potential individual and household determinants of STH infection including a survey of STH infections among household members at the time of recruitment of newborns into the cohort. Although we were able to maintain high rates of retention of the cohort during the 8 years of follow-up, fewer individuals provided stool samples (65% or more for time points with active follow-up [Fig 1]).

The collection of single stool samples may have underestimated the risk of STH infections. All children (and household members at baseline) with a positive stool sample for STH were offered anthelmintic treatment. Anthelmintic treatments given to household members would have reduced STH transmission intensity within infected households and risk of infection in cohort children during infancy. Likewise, anthelmintic treatments given to pre-school cohort children with a positive stool sample will have reduced infection frequencies and intensities. Data on most household determinants measured were collected by questionnaire around the time of the child's birth and may be subject to reporting biases. Data on some variables were collected periodically to measure time-varying effects but these did not include variables that might be linked to longitudinal STH risk such as potable water and STH among household members which were collected only at the time of recruitment into the cohort.

The dominant STH parasites in this coastal setting in Esmeraldas Province were *A. lumbricoides* and *T. trichiura* with peak prevalence between 3 and 5 years. *A. lumbricoides* emerged first while *T. trichiura* appeared later being more prevalent by 8 years of age. A decline in prevalence of STH infections at 8 years probably reflects annual treatments with single dose (400 mg) albendazole given to primary school children. Few infections with other STH parasites were observed during the first 8 years of life (1.3% for hookworm and 1.6% for *S. stercoralis*). A recent national survey of STH among schoolchildren aged 6 to 16 years in Ecuador with sample collections between 2011 and 2012 [21], estimated a lower prevalence in coastal regions of any STH of 14.5%, *A. lumbricoides* of 4.3%, and *T. trichiura* of 11.9% compared to 20.4%, 12.2%, and 13.3%, respectively, in cohort children at 8 years (S2 Table). Few hookworm infections were detected in coastal (0%) or highland (0.3%) schoolchildren, but STH infections (58.9%) including hookworm (14.6%) were more frequent (14.6%) in the Amazon region where greater poverty likely favour transmission [22]. The higher risk of infection observed

 

among schoolchildren in the Amazon is comparable to the relative frequencies observed in another region of Esmeraldas Province before 2008 [23].

Infections with *A. lumbricoides* were observed more frequently than *T. trichiura* at an earlier age but by 2 years frequencies of the two parasite species were similar, an observation also seen in a previous longitudinal study of infants in Iquitos, Peru [7]. It is possible that *A. lumbricoides* eggs—that appear to be more resilient, sticky, embryonate more rapidly, and can survive for years [24, 25]—may be more uniformly distributed in the environment increasing risk of infection in infants [4]. Opportunities for ingestion of embryonated *T. trichiura* eggs may increase as children become older and start to explore the environment beyond the confines of the immediate household. We observed strong associations between *T. trichiura* infections and living in the poorest households and with infected siblings. Sharing a poor household with infected siblings, who might be expected to contaminate frequently the peri-domestic environment with *T. trichiura* eggs, could maximize the chance of transmission to younger siblings. An analysis in an indigenous population in Panama showed spatial clustering of *T. trichiura*, but not *A. lumbricoides*, to areas with households with the greatest levels of poverty [4].

Maternal and household factors indicative of greater poverty and marginalisation were important determinants of STH infection risk as observed elsewhere [4, 8, 26–29]. These factors included maternal Afro-Ecuadorian ethnicity and lower educational level, household overcrowding, and having a latrine rather than a water closet for disposal of faeces. Having a household member infected with a STH around the time of the child's birth was a strong predictor for subsequent risk of STH infection. It didn't seem to matter so much which household member was infected, except for *T. trichiura* that was more strongly associated with having infected siblings. We identified a strong association with maternal STH in a previous analysis of STH to 3 years in this cohort [8]. Here, although strong associations were seen with maternal STH, particularly among children with mothers with greater parasite burdens for *A. lumbricoides* and *T. trichiura*, the STH variable that best fit the data for any STH and *A. lumbricoides* was STH in any household member. This observation did not seem to support an independent maternal effect and has important implications for STH control–ensuring that everyone in households with at least one infected member are treated with anthelmintic drugs will reduce the future risk of STH infections among infants and young children. Although women of reproductive age are currently included in recommendations for anthelmintic treatment in areas of high prevalence for hookworm and *T. trichiura* [30], strategies that seek to go beyond control to interruption of STH transmission likely will require community-based mass drug administrations, preferably administered at a household level by community health workers to ensure maximum coverage rates with anthelmintic drugs.

During the 12-year calendar period during which this study was conducted (2005–2017), there appeared to be a decline in the risk of infection, primarily with *T. trichiura*, in children older than 3 years–observed as a halving of the risk of infection among cohort child compared to their siblings of a similar age when sampled around the time of birth of the cohort child. In parallel with such declines in infection risk were changes in characteristics of cohort households (type of construction, overcrowding, and having adequate sanitation) coincident with a period of economic growth and increases in national investments in infrastructure and social support programmes such as conditional cash transfers (bono de desarollo humano) [21]. The rural district in which this study was done is undergoing rapid changes relating to urbanization [31]. *T. trichiura* risk seemed to be more sensitive than *A. lumbricoides* to improvements in household circumstances, suggesting perhaps that *T. trichiura* infections are better indicators of severe poverty.

We used the dataset to explore two *a priori* hypotheses: 1) zoonotic *Ascaris* spp. from pigs can be a source of human infections [32, 33]–analyses of these data did not support a greater

risk of *A. lumbricoides* (morphologically indistinguishable from pig *A. suum*) among children with at least one pig in the household compound during the course of the study; and 2) an allergic predisposition is associated with increased resistance to STH infections [34, 35] using parental history of allergic symptoms and atopy to define high-risk for allergic predisposition–analyses of these data provided some evidence of an inverse association between paternal allergic symptoms and childhood *A. lumbricoides* in age-adjusted but not multivariable analyses.

In conclusion, we followed a birth cohort for the acquisition of STH infections in a rural district in tropical coastal Ecuador and showed that approximately one half acquired infections during the first 8 years of life, almost exclusively with *A. lumbricoides* and *T. trichiura*. Individual and environmental factors indicative of greater poverty and marginalisation were associated with childhood STH infections including having an infected household member. Although maternal STH were strongly associated with infections in offspring, the effect was explained better by a shared home environment rather than a specific maternal effect. Our findings are likely to be relevant to children growing up in transitional regions in Latin America undergoing rapid changes related to urbanization. Improvements in living conditions in such populations will likely lead to significant reductions in STH infection risk. As we transition from STH control to elimination strategies in such settings, targeting of infected households for anthelmintic treatments is likely to contribute to the interruption of transmission.

## Supporting information

**S1 Table. Estimated risk of infection with soil-transmitted helminths (STH) using data from stool samples collected from 6,800 individuals living 1,973 cohort households around the time of birth of the cohort child.** Shown are estimated proportions (prop) (and 95% confidence intervals [CI]) infected by age using two-level logistic regression accounting for the household hierarchical structure of the data. *S. stercoralis*–*Strongyloides stercoralis*.
(DOCX)

**S2 Table. Raw data and predicted age-dependent risk of infections with soil-transmitted helminths in ECUAVIDA.** Predicted data were derived from models fitted using population-averaged generalized estimating equations.
(DOCX)

**S3 Table. Age-adjusted and multivariable associations between *A. lumbricoides* infection during first 8 years of life and individual, parental, and household factors including *A. lumbricoides* infections among household members.** Estimates show population-averaged estimates using generalized estimating equations. *Time-varying variables. ‡Anthelmintic treatment during the previous 12 months. N-Afro.–non-Afro-Ecuadorian; Prim.–primary completed; Second.–secondary completed; SES–socioeconomic status; overcrowding–persons/sleeping room; Income- monthly household income; Non-trad.–non-traditional (wall construction with cement/blocks); trad.–traditional (wall construction with wood/bamboo); material goods–number of household electrical goods; pigs–keeping pigs around the house; agriculture–child lives on a farm or visits a farm at least once a week; NEG- negative; MOD-HEAVY–moderate and heavy intensity infections.
(DOCX)

**S4 Table. Age-adjusted and multivariable associations between *T. trichiura* infections during first 8 years of life and individual, parental, and household factors including *T. trichiura* infections among household members.** Estimates show population-averaged estimates using generalized estimating equations. *Time-varying variables. ‡Anthelmintic treatment during the previous 12 months. N-Afro.–non-Afro-Ecuadorian; Prim.–primary

completed; Second.–secondary completed; SES–socioeconomic status; overcrowding–persons/sleeping room; Income- monthly household income; Non-trad.–non-traditional (wall construction with cement/blocks); trad.–traditional (wall construction with wood/bamboo); material goods–number of household electrical goods; pigs–keeping pigs around the house; agriculture–child lives on a farm or visits a farm at least once a week; NEG- negative; MOD-HEAVY–moderate and heavy intensity infections.
(DOCX)

**S5 Table. Age-adjusted associations between *A*. *lumbricoides* infection intensity during first 8 years of life and individual, parental, and household factors including *A*. *lumbricoides* infections among household members.** *A*. *lumbricoides* infection intensity was measured as eggs per gramme (epg) of stool. Original epg counts were fit using a zero-inflated model. Zeros model represents associations of a positive count with variables (denoted by 1 in our previous analyses) (OR>1 indicates a positive association with positive counts while <1 indicates an association with zero counts). Counts model represents associations using the negative binomial distribution (Fold-change>1 indicates fold increase in egg counts associated with that variable while <1 indicates corresponding fold decrease).
(DOCX)

**S6 Table. Age-adjusted associations between *T*. *trichiura* infection intensity during first 8 years of life and individual, parental, and household factors including *T*. *trichiura* infections among household members.** *T*. *trichiura* infection intensity was measured as eggs per gramme (epg) of stool. Original epg counts were fit using a zero-inflated model. Zeros model represents associations of a positive count with variables (denoted by 1 in our previous analyses) (OR>1 indicates a positive association with positive counts while <1 indicates an association with zero counts). Counts model represents associations using the negative binomial distribution (Fold-change>1 indicates fold increase in egg counts associated with that variable while <1 indicates corresponding fold decrease).
(DOCX)

**S7 Table. Estimates for associations between any childhood soil-transmitted helminth (STH) infection and individual, parental, and household determinants derived using generalized estimation equations (GEE) and Maximum Likelihood Estimates (MLE) for binary longitudinal outcomes under missing completely at random and missing at random assumptions, respectively.** GEE provide population average estimates while MLE provide subject-specific estimates.
(DOCX)

**S8 Table. Sensitivity analyses for associations between any childhood soil-transmitted helminth (STH) infections and individual, parental, and household determinants derived using generalized estimation equations (GEE).** Complete data analyses from Table 1 are provided for comparison. All ORs were adjusted for age.
(DOCX)

**S1 Data. Raw data used for analyses**
(TXT)

## Acknowledgments

We thank the ECUAVIDA study team for their dedicated work and the cohort mothers and children for their enthusiastic participation. We acknowledge the support of the Ecuadorian

Ministry of Public Health and the Directors and Staff of the Hospital "Padre Alberto Buffoni", Quinindé, Esmeraldas Province.

## Author Contributions

**Conceptualization:** Philip J. Cooper.

**Data curation:** Martha E. Chico, Philip J. Cooper.

**Formal analysis:** Irina Chis Ster, Hamzah F. Niaz.

**Funding acquisition:** Philip J. Cooper.

**Investigation:** Irina Chis Ster, Martha E. Chico, Yisela Oviedo, Maritza Vaca, Philip J. Cooper.

**Methodology:** Irina Chis Ster, Philip J. Cooper.

**Project administration:** Martha E. Chico, Philip J. Cooper.

**Resources:** Philip J. Cooper.

**Supervision:** Martha E. Chico, Philip J. Cooper.

**Writing – original draft:** Irina Chis Ster, Hamzah F. Niaz, Philip J. Cooper.

**Writing – review & editing:** Martha E. Chico, Yisela Oviedo, Maritza Vaca.

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
