## [Decision Letter · Decision Letter 0]

17 Aug 2021

Dear Dr. Cooper,

Thank you very much for submitting your manuscript "The epidemiology of soil-transmitted helminth infections in children up to 8 years of age: findings from an Ecuadorian birth cohort" for consideration at PLOS Neglected Tropical Diseases. As with all papers reviewed by the journal, your manuscript was reviewed by members of the editorial board and by several independent reviewers. The reviewers appreciated the attention to an important topic. Based on the reviews, we are likely to accept this manuscript for publication, providing that you modify the manuscript according to the review recommendations. 

Sincerely,

Darren J. Gray

Associate Editor

Aaron Jex

Deputy Editor

Reviewer's Responses to Questions

**Key Review Criteria Required for Acceptance?**

**Methods**

-Are the objectives of the study clearly articulated with a clear testable hypothesis stated?

-Is the study design appropriate to address the stated objectives?

-Is the population clearly described and appropriate for the hypothesis being tested?

-Is the sample size sufficient to ensure adequate power to address the hypothesis being tested?

-Were correct statistical analysis used to support conclusions?

-Are there concerns about ethical or regulatory requirements being met?

Reviewer #1: Yes

Reviewer #2: This MS describes a very impressive prospective study, an approach that is rare as highlighted by the authors. Particular strengths include the length of the longitudinal study, the large sample size, the fact that 4 diagnostic methods were used and the complex statistical analyses employed.

**Results**

-Does the analysis presented match the analysis plan?

-Are the results clearly and completely presented?

-Are the figures (Tables, Images) of sufficient quality for clarity?

Reviewer #1: Yes

Reviewer #2: My comments are minor and mainly relate to some more explanation of the statistical analyses.

Abstract

Line 37: “Peaked in risk” sounds a little clumsy to me

Methods

I would like to see a little explanation as to why the authors decided not use intensity (e.p.g) as a continuous variable in their analyses?

Line 157: I would like to see an explanation of “two level logistic regression”

Line 170: Binary random effects models were also considered – with what decision?

Line 188: Patterns in missing data analyses and sensitivity analyses – again what was the outcome of these analyses? Could the results in the supplementary table be summarised in some way in the main body of the MS to inform readers more easily? I know you mention this in the Discussion but I think a little more detail is required.

Line 201: What is “minor” assent?

**Conclusions**

-Are the conclusions supported by the data presented?

-Are the limitations of analysis clearly described?

-Do the authors discuss how these data can be helpful to advance our understanding of the topic under study?

-Is public health relevance addressed?

Reviewer #1: Yes

Reviewer #2: Figure 2: There are interesting differences in the degree of infection between mothers and fathers. Could you provide possible explanations?

It is very interesting to me that you observe differences in Ascaris and Trichuris as too often, these two parasites are conflated epidemiologically. Can you address the issue of their differences a little more in your Discussion for example differences egg survival, transmission pattern etc?

**Editorial and Data Presentation Modifications?**

Reviewer #1: None

Reviewer #2: See above

Minor revision only

**Summary and General Comments**

Reviewer #1: The work by Ster et al focusses on a large Ecuadorian birth cohort followed from birth to the age of 8 in the context of soil transmitted helminths. As far as I am aware, this is a pretty unique cohort in terms of the length of study (8 years), the data from which offers to identify risk factors for STH infections. This sort of data is of immense value to the research community and the comparisons made range across for example age, day-care, educational level, and infection in other household members. The authors conclude, perhaps not surprisingly, that STH infections among the house hold members is a key risk factor; but there are also other important gains of knowledge within this manuscript too.

Some interesting differences in the context of risk factors for having Ascaris versus Trichuris are identified, which would be worth discussing. For example:

Why would having a WC inversely associate with Ascaris but not whipworm? 

Does the lack of an inverse association between whipworm and drug treatment infer drug resistance within Trichuris parasite population?

Why do Ascaris infections emerge first before whipworm? 

Why for Trichuris does having infected sibling matter so much more re risk of subsequent STH infection? 

Further would it also be possible from the data sets to analyse whether attendance at school altered risk? (at what age do children start attending school?)

Overall the manuscript provides new important knowledge which will be of interest to the readers of PLOS NTDs

Reviewer #2: (No Response)

PLOS authors have the option to publish the peer review history of their article (what does this mean?). If published, this will include your full peer review and any attached files.

Reviewer #1: Yes: Kathryn Else

Reviewer #2: No

Figure Files:

Data Requirements:

Reproducibility:

References

---

## [Editor Report · Decision Letter 1]

3 Nov 2021

Dear Dr. Cooper,

We are pleased to inform you that your manuscript 'The epidemiology of soil-transmitted helminth infections in children up to 8 years of age: findings from an Ecuadorian birth cohort' has been provisionally accepted for publication in PLOS Neglected Tropical Diseases.

Best regards,

Aaron R. Jex

Deputy Editor

Aaron Jex

Deputy Editor

---

## [Editor Report · Acceptance letter]

16 Nov 2021

Dear Dr. Cooper,

We are delighted to inform you that your manuscript, "The epidemiology of soil-transmitted helminth infections in children up to 8 years of age: findings from an Ecuadorian birth cohort," has been formally accepted for publication in PLOS Neglected Tropical Diseases.

Best regards,

Shaden Kamhawi

co-Editor-in-Chief

Paul Brindley

co-Editor-in-Chief
